# Clinical Effects of Immuno-Oncology Therapy on Glioblastoma Patients: A Systematic Review

**DOI:** 10.3390/brainsci13020159

**Published:** 2023-01-17

**Authors:** Masoumeh Najafi, Amin Jahanbakhshi, Sebastiano Finocchi Ghersi, Lucia Giaccherini, Andrea Botti, Francesco Cavallieri, Jessica Rossi, Federico Iori, Cinzia Iotti, Patrizia Ciammella, Mohsen Nabiuni, Marzieh Gomar, Omid Rezaie, Salvatore Cozzi

**Affiliations:** 1Skull Base Research Center, Iran University of Medical Sciences, Tehran 1997667665, Iran; 2Stem Cell and Regenerative Medicine Research Center, Iran University of Medical Sciences, Tehran 1997667665, Iran; 3Radiation Oncolgy Unit, AOU Sant’Andrea, Facoltà di Medicina e Psicologia, Università La Sapienza, 00185 Rome, Italy; 4Radiation Oncology Unit, Azienda USL-IRCCS di Reggio Emilia, 42123 Reggio Emilia, Italy; 5Medical Physics Unit, Azienda USL-IRCCS di Reggio Emilia, 42123 Reggio Emilia, Italy; 6Neurology Unit, Neuromotor & Rehabilitation Department, Azienda USL-IRCCS di Reggio Emilia, 42123 Reggio Emilia, Italy; 7Clinical and Experimental Medicine PhD Program, University of Modena and Reggio Emilia, 41125 Modena, Italy; 8Department of Neurosurgery, Iran Univesity of Medical Sciences, Tehran 1997667665, Iran; 9Radiation Oncology Research Center, Iran Cancer Institute, Tehran University of Medical Sciences, Tehran 1416753955, Iran; 10Hematology-Oncology Department, Jam Hospital, Tehran 1997667665, Iran; 11Radiation Oncology Deptartement, Centre Léon Bérard, 69373 Lyon, France

**Keywords:** immunotherapy, chemotherapy, radiotherapy, glioblastoma, GBM, temozolomide, immuno-oncologic therapy, cell therapy, vaccine, vaccination, cell-based therapy, check-point inhibitor, oncolytic viral therapy

## Abstract

The most prevalent and deadly primary malignant glioma in adults is glioblastoma (GBM), which has a median survival time of about 15 months. Despite the standard of care for glioblastoma, which includes gross total resection, high-dose radiation, and temozolomide chemotherapy, this tumor is still one of the most aggressive and difficult to treat. So, it is critical to find more potent therapies that can help glioblastoma patients have better clinical outcomes. Additionally, the prognosis for recurring malignant gliomas is poor, necessitating the need for innovative therapeutics. Immunotherapy is a rather new treatment for glioblastoma and its effects are not well studied when it is combined with standard chemoradiation therapy. We conducted this study to evaluate different glioblastoma immunotherapy approaches in terms of feasibility, efficacy, and safety. We conducted a computer-assisted literature search of electronic databases for essays that are unique, involve either prospective or retrospective research, and are entirely written and published in English. We examined both observational data and randomized clinical trials. Eighteen studies met the criteria for inclusion. In conclusion, combining immunotherapy with radiochemotherapy and tumor removal is generally possible and safe, and rather effective in the prolongation of survival measures.

## 1. Introduction

The most frequent primary malignant brain tumor is glioblastoma (GBM). It has a median survival time of just 15 months, despite stringent clinical standards of treatment that include gross total resection, high-dose radiation therapy, dose-intensified temozolomide chemotherapy, and tumor-treating fields (TTF) [1,2]. There is an urgent need for new treatments, and immunotherapy has recently become very promising for treating cancer in different settings [3,4,5,6]. There are several modalities for immunotherapy. Recent clinical studies have shown that immune checkpoint suppression with monoclonal antibody-based blocking of the programmed cell death protein 1 (PD1) and its ligand can have clinical and radiographic effects in certain patients with advanced malignancies. After the Food and Drug Administration approved Sipleucel T, an autologous cellular vaccine that increases survival for patients with advanced castration-resistant prostate cancer, these clinically significant immunotherapies were approved [7]. In people with a variety of solid and hematologic malignancies, vaccination with irradiated autologous tumor cells that are modified to produce granulocyte–macrophage colony-stimulating factor (GM-CSF), or “GVAX”, has triggered a robust antitumor immunity and extended life in some patients [8]. A new therapy option being investigated in preclinical studies and clinical trials is dendritic cell (DC) immunization [8,9,10,11]. DC vaccination seeks to stimulate the patient’s immune system against the tumor because DCs are the most effective antigen-presenting cells. Additionally, establishing long-term antitumor protection may be possible through the development of immunological memory.

Adopting immunotherapy treatments for other cancers to glioblastoma is possible, and different studies have documented some benefits [6,12,13]. Other researchers have previously shown that 90 percent of GBMs have cytomegalovirus (CMV) antigens, although the normal brain does not [14,15,16]. Even though DC-based immunity treatment for high-grade gliomas (HGGs) appears to be effective, its therapeutic benefit may only apply to a small subset of patients. It is persistently seen as a tail in the overall survival (OS) curve in survival assessments of patients who received vaccinations [13,17,18,19]. Vaccinations against the heat-shock protein (HSP) promote antigen absorption by antigen-presenting cells and activated T lymphocytes [20]. It triggers an immune response against the tumor that is both innate and adaptive [21].

A combination of immunotherapy and radiotherapy may have some synergistic effects in terms of efficacy and complications. Moreover, the combination of immune radiation could modulate the microenvironment, rendering it better at tumor killing by priming the quiescent host immune system [22]. Patients with glioblastoma may develop inflammatory lesions following chemotherapy and radiation (called “pseudo-progression”). Since their effect is to trigger an inflammatory response against the tumor, vaccinations and other immunotherapies may worsen this issue. It is critical to accurately distinguish between lesions brought on by treatment and tumor growth to prevent pointless procedures and the suspension of potentially curative treatments. Guidelines for immunotherapy response assessment in neuro-oncology (iRANO) were just released [23,24]. Patients with glioblastoma generally have a short progression-free survival period, and the formation of new lesions or progression occurs soon in the course of the disease. However, this may not necessarily accompany clinical deterioration and is not a definite sign of therapy failure in immunotherapy. The GBM’s dismal survival rate may interfere with research on immunotherapy treatments because they usually need some time to show their maximum effect. Interpretation of immunochemoradiation therapy in GBM should take these problems into account [25].

However, some authors warn of possible immune-related adverse events and a synergistic effect of radiotherapy and immunotherapy on toxicities [26]. Therefore, elucidating a combination of immunotherapy and radiotherapy for glioblastoma in the existing literature is a hot research topic in terms of feasibility, effectiveness, and safety.

## 2. Materials and Methods

### 2.1. Search Strategy

A thorough review of the literature served as the study design for this project. To publish systematic reviews of research evaluating health care treatments, this review adhered to the Preferred Publishing Items for Systematic Reviews publication guideline [27]. The review was generated by the following question: What is the clinical impact of immunotherapy in the management of glioblastoma?

In addition to using the papers found through electronic search and review, this was accomplished by carefully searching the most significant and pertinent medical resource databases, including PUBMED, SCOPUS, Cochrane, and Science Direct between 2015 and 2022, based on the considered keywords, including glioblastoma, glioblastoma multiforme OR GBM, AND immune therapy, immunotherapy, immune cell therapy, immune-oncologic therapy, cell therapy, vaccine, vaccination, cell-based therapy, checkpoint inhibitor OR oncolytic viral therapy. Their sources were investigated, a manual search was conducted, and, if necessary, a discussion with subject-matter specialists was coordinated. The proper search term (Mesh, Free text) was used.

### 2.2. Study Selection and Inclusion/Exclusion Criteria

Two blinded investigators searched databases for all eligible studies. Eligibility was determined using the title or abstract and, if necessary, the whole content. This systematic review evaluated the clinical impact of immuno-oncology in patients with glioblastoma. The essay also needed to be original, based on recent English-language publications, and either prospective or retrospective research to qualify for inclusion. The researchers examined both observational data and randomized clinical trials. Exclusion criteria were lack of access to the full text of the manuscript; studies with unclear or irreproducible results (i.e., lack of clear outcomes or presence of errors in methodology and/or analyses); and review papers.

### 2.3. Data Extraction

Two unblinded reviewers independently performed the data collection on structured collection forms. We resolved disagreements by consensus or by involving a third person.

## 3. Results

According to the purpose of the study, the initial search returned over 750 studies with pertinent information using keywords and references. After titles and abstracts were checked, complete texts were examined, and 18 papers were found to meet the criteria for admission, with a total number of 1025 patients (Figure 1).

The studies were phase I and II studies and evaluated the use of immunotherapy in both newly diagnosed and recurrent glioblastoma. Table 1 summarizes the salient features and conclusions of the eighteen studies that were included.

To summarize these data, they can be classified according to the information they present in terms of feasibility, efficacy, and safety of immunotherapy in glioblastoma patients. These studies were phase I or phase II trials that were nonhomogenous regarding the patient characteristics, tumor condition, treatment modality, and outcome measures. Therefore, a pooled analysis could not be performed.

Feasibility

Vaccine-based immunotherapy composed the most prevalent strategy in the selected studies. Dendritic cell vaccination (DCV) was used in five studies and other vaccination strategies were used in four. Batich et al. used CMV-specific DCV while the standard glioblastoma regime (chemoradiation) was maintained. They pulsed DCs with Td toxoids that improved lymph node migration and outcomes [28]. Akasaki et al. fused autologous glioma cells with autologous DCs using polyethylene glycol and injected the product intradermally [29]. Antonopoulos et al. found that DCV concurrent with temozolomide and delayed DCV are both feasible and effective [30]. Hsu et al. conducted their DCV by extracting DCs from peripheral blood and pulsing them with tumor lysate overnight before injection into the patients [31]. Inoges et al. used the same approach and found it sufficient to produce at least six vaccine doses [9]. Desjardins et al. used a convection-enhanced delivery system to inject a nonpathogenic polio/rhinovirus (PVSRIPO) chimera into the tumor resection cavity. The safe delivery of 10^7^ to 10^10^ 50% tissue-culture infectious doses (TCID50) was possible [32]. Oji et al. used the Wilms Tumor (WT)-1 intradermal vaccination for recurrent GBM cases and efficiently found both cellular and humoral responses [33]. Curry et al. mixed irradiated homologous glioma cells with irradiated GM-K562 cells and were able to obtain sufficient cells in 10 of 11 patients for whom at least four vaccinations with doses 5 × 10^6^ to 1 × 10^7^ were possible [34].

The other common strategy is T-cell-based therapy. Brown et al. could produce a sufficient number of CAR T cells against IL13Rα2 and used it for the first time in three patients, delivered intracranially using a catheter-reservoir system [35]. Later, they used up to 10 × 10^6^ per infusion to treat a recurrent multifocal and intraspinal GBM patient. The injection was performed several times into the tumor resection cavity and cerebral ventricles [36]. Guo et al. produced a chimeric switch receptor T cell (CSR-T) via chimerization of the extracellular part of PD-1 and the intracellular part of CD28. They could detect the increased levels of IFN-gamma, IL-6, and T cells following intraventricular or intravenous administration of 10^8^ CSR-T cells [37]. O’Rourke et al. synthesized EGFRvIII-directed CAR T cells for the first time in ten recurrent GBM cases. They revealed the transient expansion of cells in all patients. the median transduction efficiency was 19.75% [38]. Kirkin et al. produced DNA-demethylated T-helper cells using dendritic cells to activate T helpers in the peripheral blood, followed by the induction of CT antigen in them. Then, they used these THs to induce up to 6.8 × 10^7^ cytotoxic T cells and treat 25 patients with recurrent gliomas [39]. In Weathers et al.’s work, temozolomide was used for lymphodepletion to maximize the expansion of the T-cell clone against the CMV antigen. In their experience, 26% failure in the expansion was observed. This failure was not seen in normal people, highlighting the background lymphopenic state in GBM patients [40]. Reap et al. also used CMV-activated T cells with or without DCV in 22 patients with newly diagnosed GBM and showed a better cytokine response in the DCV group [40,41].

Checkpoint inhibitors are the third common immunotherapy strategy. Duerinck et al. used intracerebral ipilimumab and nivolumab plus intravenous nivolumab in 27 patients. they administered iv nivolumab before and after surgery and injected both drugs into the tumor resection cavity [42]. George et al. introduced a study using durvalumab in 162 patients [43]. In clinical trials, PD-1 inhibition via nivolumab, concurrently with chemoradiotherapy or radiation therapy, did not improve progression-free survival (PFS) or OS in newly diagnosed glioblastoma [44]. Similarly, it did not improve OS compared with bevacizumab in recurrent glioblastoma. Although the response rate to nivolumab was low (8%) in patients with recurrent glioblastoma, the response was more durable relative to bevacizumab [45]. Cloughesy et al. compared the neoadjuvant plus adjuvant treatment of recurrent glioblastoma patients with pembrolizumab to just adjuvant treatment with this PD-1 blocking drug. They found that in the neoadjuvant group, patients had significantly higher overall survival. Therefore, neoadjuvant treatment should be considered for future studies [45].

Efficacy

Most studies do not give sufficient information about the efficacy and the effect on overall survival and progression-free survival because they are often single-armed and lack enough statistical power to be compared with historical cohorts. Moreover, the patient characteristics are so different that they could not be matched in a comparison. Some studies with a limited number of cases reported long-term survivors. The survival rates are presented in Table 1. Duerinck et al. reported a statistically significant longer overall survival (median: 38 months) after treatment with nivolumab and ipilimumab in comparison with matched historical cohorts [41]. Other studies have reported overall survival rates which are higher relative to the standard treatment and have also reported multiple cases with exceptionally long survivals. However, the observed effect cannot be tested by a perfect statistical design [8,28,31,32,34,35,38]. Batich reported more than 5-year survival times in one-third of patients, irrespective of age, performance status, IDH, and MGMT [27].

Oji et al. deduced that when IgG (humoral response) is increased in response to WT-1 peptide vaccination, in addition to the cellular response, it is strongly associated with a higher PFS and OS [32]. Hsu et al. mentioned that the tumor-infiltrating lymphocytes (TILs) are associated with time-to-progression (TTP), but just in patients who received DCV and not in control patients [30]. Antonopoulos et al. showed that there is no difference in the overall survival of patients who received DCV either concurrently or after temozolomide chemotherapy [29].

Safety

In terms of adverse effects and toxicities, the results are rather consistent. No study reported a high incidence of serious complications. No systemic autoimmune reaction was reported. According to the grading system proposed by the National Cancer Institute Common Terminology Criteria for Adverse Events, grade I or II complications such as fever, neutropenia, and thrombocytopenia were fairly common. However, despite concomitant chemoradiation and immunotherapy in most studies, grade III and higher complications that could be attributed to the treatment regimen were a rare occurrence.

In the study by Desjardins et al., of nine patients in the dose-escalation phase, two had grade III adverse events, and of fifty-two patients in the dose-expansion phase, nine patients experienced grade III adverse events and one had a grade V adverse event attributable to PVSRIPO [31]. Inoges et al. reported two neutropenia and thrombocytopenia grade III events in their study that were attributable to chemotherapy [8]. Weathers et al. also encountered two grade IV neutropenia and thrombocytopenia and one grade III seizure and confusion [39]. In the study by Brown et al., of three patients, one had grade III headaches and one had grade III transient neurologic deficits when treated with 10^8^ doses, but not with 5 × 10^7^ doses [34]. In the other study using EGFRvIII CAR T cells, two patients, for whom silituximab (anti-IL-6) was administered, developed seizures and neurologic decline [37]. There were no sufficient data to compare complications in different types of immunotherapies.

Concerning radiologic lesions that may occur after treatment with chemoradiation or immunotherapy, it is difficult to distinguish them from each other or from tumor progression. There might be a synergistic effect between different treatment modalities, and there may be a significant time overlap between tumor progression or treatment response (pseudoprogression). Alcaide-Leon et al. found a higher incidence of treatment-related lesions in newly diagnosed GBM cases rather than recurrent ones. They used ADC (apparent diffusion coefficient) imaging for the diagnosis [46].

**Table 1 brainsci-13-00159-t001:** The primary features and outcomes of the selected studies.

Ref.	Type of Phase	Mean Age	Case of Glioblastoma (Male-Female)	Patients Characteristics	Immunotherapy	Conclusions
Reap EA et al. (2021) [41]	Randomized phase II	56.5	17 patients (12-5)	Newly diagnosed GBM	Vaccination with cytomegalovirus phosphoprotein 65(pp65) RNA-loaded dendritic cells	No severe adverse events. Polyfunctional T-cell responses correlated with improved survival.
George et al. (2022) [43]	Phase II	55.2	162 patients (69%, 31%)	Newly diagnosed GBM		Median PFS: 106 days. Median OS: 207.5 days.
Antonopoulos et al. (2019) [30]	Phase II	58	101 patients (65-36)	Newly diagnosed high-grade gliomas	Dendritic cell vaccination during adjuvant temozolomide (TMZ)	2-year OS: 33.6%. Median OS:19 months. No difference in OS for patients treated with vaccine. Concurrent or adjuvant TMZ. Specific immune profiles may predict the result.
Curry et al. (2012) [34]	Phase I trial	53.34	10 patients (9-1)	Recurrent GBM	Granulocyte-macrophage colony-K562 cells mixed with autologous glioma cells	No dose-limiting toxicity. A clear increase in the intensity of inflammatory infiltrates. Increased activation markers in both CD_4_^+^ and CD_8_^+^ T cells. Increased antibody response to angiogenic cytokines.
Weathers at al. (2020) [40]	Phase I/II	50.1	20 patients	Newly diagnosed GBM	Cytomegalovirus pp65-specific T cells	No treatment-related toxicity. Median PFS: 1.3 months. Median OS: 12 months. Seropositivity for CMV does not necessarily means a good tumor response to treatment. Repetitive injections reduced cytokines and effector activity of T cells, probably because of antigen heterogeneity.
Brown et al. (2015) [35]	Phase I study	48.5	3 patients	Recurrent GBM	IL13(E13Y)-zetakine CD8 CTL targeting IL13Ra2	First-in-human trial of IL13Ra2-specific CAR T cells. Response in 2 of 3 patients in terms of reduced IL-13 expression and reduction in MRI lesion.
Guo et al. (2019) [37]	Phase I study	44	14 patients	Recurrent GBM	Chimeric switch receptor T (CSR-T) cells	No grade III or greater adverse events. The median OS: 4.40 months. Increased levels of IFN-gamma and IL-6, and increased number of T cells in CSF.
Batich et al. (2020) [28]	Randomized Phase II study	50.5	50 patients.	Newly diagnosed GBM	Cytomegalovirus (CMV)-specific DC vaccines	Nearly one-third of patients were long-term survivors (survived > 5 years). This effect was independent of age, KPS, and IDH or MGMT mutations
Kirkin et al. (2018) [39]	Phase I study	-	25 patients	Recurrent GBM	Activation of normal lymphocytes with DNA-demethylated T-helper cells	No treatment-related adverse effects. A large number of CD8+ and NK cells were detected. Disease control in 5 of 10 patients receiving 3 injections. In 3 patients: long-term regression of the tumor.
Akasaki et al. (2016) [29]	Phase I/II study	54.6	32 patients	Both newly diagnosed and recurrent GBM	Fusion of autologous DCs and glioma cells using polyethylene glycol	No grade ≥ 3 toxicity. Medians of PFS and OS of Group-Recurrence (n = 10) were 10.3 and 18.0 months, and those of Group-Newly diagnosed (n = 22) were 18.3 and 30.5 months, respectively.
Alcaide-Leon et al. (2020) [47]	Phase II study	55	27 patients (15-12)	Newly diagnosed GBM	Heat-shock protein (HSP) vaccine	The iRANO advice of a 6-month window in which worsening illness should not be notified after immunotherapy begins was not supported by this study.
Duerinck at al. (2021) [42]	Phase II study	55	27 patients with recurrent GBM (17-10)	Recurrent GBM	Ipilimumab (IPI) and nivolumab (NIVO)	Median OS: 38 weeks. 6-month, 1-year, and 2-year OS rates: 74.1%, 40.7%, and 27%, respectively.
Oji et al. (2016) [33]	Phase II study	51	59 patients (37-22)	Recurrent GBM	Wilms’ tumor gene 1 (WT1) peptide vaccination	Median PFS: 83 days. Median OS: 252 days. Humoral response (Ig-G production) against WT-1 associated with longer PFS and OS. Longest survival in patients who developed both IgG response and delayed-type hypersensitivity.
Desjardins et al. (2018) [32]	Phase II study	55	61 patients (36-25)	Recurrent GBM	Recombinant nonpathogenic polio–rhinovirus chimera (PVSRIPO)	At 24 and 36 months, the survival rate for patients who underwent PVSRIPO immunotherapy was greater than the rate for historical controls. Two patients remained alive for 69 months.
Hsu et al. (2016) [31]	Phase I/II study		15 patients	Newly diagnosed GBM	Dendritic cells pulsed with tumor lysate overnight	Time to progression (TTP) and overall survival significantly correlated with tumor-infiltrating lymphocyte (TIL) concentration.
Brown et al. (2016) [36]		50	A 50-year-old man	Recurrent GBM	Chimeric antigen receptor (CAR)-engineered T cells	Multiple intracranial and spinal tumors disappeared according to MRI and PET after CAR T-cell therapy, and this response maintained for 7.5 months.
O’Rourke et al. 2017) [38]	Phase I study	60.5	10 patients (5-5)	Recurrent GBM	EGFRvIII-directed CAR T cells	No dose-limiting toxicity. The first trial in humans. In all patients, transient expansion of CAR T cells was detected in peripheral blood transduction. Efficiency: 19.75%. Target dose: 1–5 × 10^8^.
Inogés et al. (2017) [9]	Phase II study	61	32 patients	Newly diagnosed GBM	Vaccination with autologous dendritic cells pulsed with whole-tumor lysate	No adverse effects related to immunotherapy. Median OS: 23.4 months. Increased immune cell proliferation and cytokine production in 11 of 27 evaluated patients. This increment did not correlate with the overall survival rate

## 4. Discussion

Despite advances in the understanding of molecular changes in glioblastoma, effective targeted therapies are lacking. Bevacizumab, a monoclonal antibody against the vascular endothelial growth factor, is the only approved addition to recurrent glioblastoma management [48]. Immunotherapy has made fundamental changes in the treatment and outcome of some cancers, such as melanoma. These changes encourage researchers to find a new treatment horizon for other cancers. Glioblastoma has been investigated for the feasibility of different immunotherapy approaches, i.e., checkpoint inhibitors, antigen-based and dendritic cell vaccination, and T-cell-based immunotherapies. Moreover, a combination of immunotherapy approaches has been tried. For example, CMV-activated T cells, when administered along with dendritic cell vaccination, may produce a higher immunologic response [41].

Checkpoint inhibitors are among the most straightforward approaches and are being used widely for other cancer types. Preclinical studies of programmed cell death protein 1 (PD-1) pathway inhibition showed promising results in gliomas [49,50]. The timing of the administration of checkpoint inhibitors may be an issue in their effectiveness. As it has been shown, adding neoadjuvant treatment may increase overall survival [46]. Among the different modalities for immunotherapy in glioblastoma, dendritic cell vaccination has gathered considerable attention after some encouraging reports showing acceptable efficacy and safety levels [13]. It is well acknowledged that DC immunization improves OS in GBM patients [18,19]. T-cell-based approaches are technically more demanding and probably more expensive. T cells can be engineered to increase the immune response nonselectively [37,39] or be modified to target specific targets [35,36,38,40].

Apart from choosing the immunotherapy modality (either of three modalities or their combinations), the approach of delivery is an important concern, especially in brain tumors because of the presence of a blood–brain barrier that limits the access of many therapeutics to the tumor microenvironment. Among the selected trials, it is obvious that besides intravenous and intradermal delivery methods, intracranial infusions have led to favorable results. Direct infusion into the tumor resection cavity or the cerebral ventricles may increase the efficiency of the technique in reducing adverse events, although it has accompanying complications specific to neurosurgical procedures, such as hemorrhage into the catheter passage canal [35,36,37,42].

Another feasibility issue is that, for most patients, particularly newly diagnosed glioblastoma patients, we cannot ignore the standard treatment for immunotherapy. In the selected trials, no patient had been deprived of radiation and temozolomide radiotherapy. Despite that, we observed sufficient efficacy and an acceptable toxicity profile.

Immunotherapy for glioblastoma is still in its childhood. So, speaking about the effectiveness is difficult due to the sparsity of well-designed trials with a sufficient number of patients. Most studies are single-armed and are very heterogenous in their patient population and treatment technique. So, the comparison of different studies with each other or with historical cohorts with standard treatment is difficult. Numerous factors affect treatment response and survival rates. It has already been proposed that immune cell subpopulations, their relative numbers, and their evolution may affect the prognosis of patients with cancer [51,52,53]. The OS and progression-free survival (PFS) of GBM patients undergoing autologous dendritic cell/tumor antigen vaccine (ADCTA) treatment are both significantly impacted by the PD-1+/CD8+ ratio. Age, the extent of the gross total tumor removal, the use of full concurrent radiation and chemotherapy (CCRT), and the PD-1 lymphocyte count are additional relevant statistical variables. Peripheral blood mononuclear cells are often known as TIL (PBMCs) [54]. So, we can just rely on the case series and small population studies that have revealed some promising outcomes (Table 1). Interestingly, there might be some synergistic effects between standard treatment and immunotherapy. There is evidence that radiation has impacts beyond just getting rid of the tumor cells with the highest radiosensitivity. Local radiation increases the vulnerability of solid tumors to immune-mediated killing, possibly through promoting dendritic cell and effector T-cell infiltration and activity [55]. The delivery of concomitant chemoradiotherapy is likely the cause of the greater prevalence of treatment-induced lesions observed in vaccination studies for newly diagnosed gliomas compared to studies in recurrent gliomas [46]. So, it seems to be a double-edged sword.

Several investigations on glioma patients who received vaccination therapy looked into the emergence of treatment-induced lesions. In a dendritic cell vaccination study, adults with recurrent gliomas were shown to have a 4.5 percent incidence [55,56]. Of the other research work, 33% experienced treatment-induced lesions. However, in our study, it is clear that different modalities of immunotherapy in combination with standard treatments are safe enough. For nontoxic doses, an adequate treatment effect has been observed in many patients. However, phase III trials should be released with larger populations and longer follow-ups to reach a more accurate safety profile.

## 5. Conclusions

The need to identify new treatments for patients suffering from primary or recurrent glioblastoma has aroused increasing interest in the use of immunotherapy. There is a huge heterogeneity in immunotherapy studies, so it is difficult to deduce a final decision about the selection of treatment protocol. In this study, we summarized the most salient immunotherapy trials on glioblastoma and tried to explain what can be extracted from them in terms of the feasibility of different treatment approaches and combinations, their efficacy, and safety. Although they have not entered clinical practice due to the small number of patients enrolled in the clinical trials, immunotherapy is safe with or without concurrent standard treatment and appears to impact overall survival. However, as it was mentioned earlier, 65% of oncologic studies are not consistent when repeated and only 6% are expected to be reproducible [27]. This problem is more evident in glioblastoma, which is a rare malignancy. The heterogenic antigen profile in glioblastoma is also an important factor that affects the reproducibility of immunotherapy in different patient populations. Therefore, studies with larger numbers of patients and better designs are needed to draw more confident conclusions.

## Figures and Tables

**Figure 1 brainsci-13-00159-f001:**
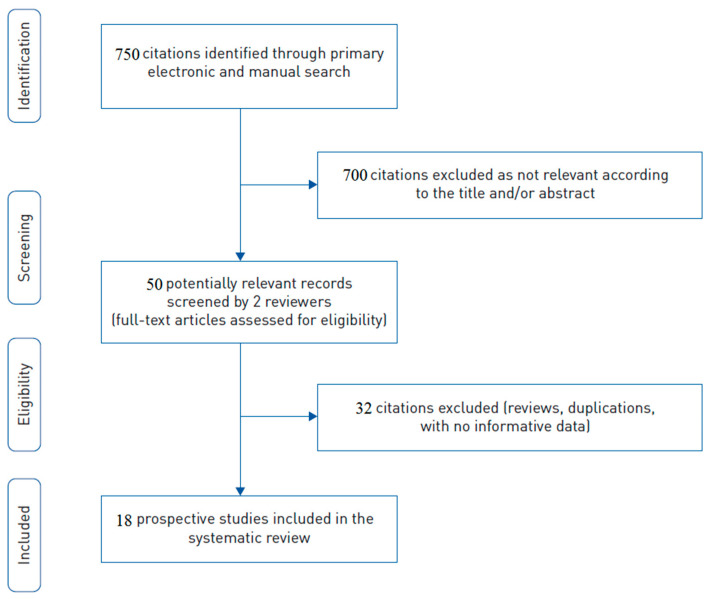
The study selection flow chart.

## Data Availability

Not applicable.

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
