# Peer review of "Clinical Effects of Immuno-Oncology Therapy on Glioblastoma Patients: A Systematic Review"

_brainsci, 2023, doi:10.3390/brainsci13020159_

Round 1
Reviewer 1 Report
This manuscript is a Systematic Review on clinical effects of immuno-oncology therapy on glioblastoma patients.
The authors selected 18 studies among 750, and they draw the conclusions that immuno-oncology treatments have been shown to be safe with or without concurrent standard treatment, without increased toxicity, and appear to impact overall survival.
Actually, I do not find well performed this manuscript. The figure inserted is really insufficient to describe the whole story. The legend is almost absent. The table is a repetition of results. The discussion is partly of interest, but really superficial. Of course, the lacking of an adequate introduction explaining several of the molecular mechanisms then discussed in the Discussion section is a minus that strongly and negatively affect the Discussion section itself.
There are numerous words with hyphen without any specific reason.
Please correct throughout the manuscript such as lines 43,44, 57,59, 80, 97, 91 and many others.
Author Response
To Reviewer 1
Dear Dr/Prof
Thank you for your nice comments. The manuscript was revised entirely according to your comment.
All sections including introduction, results, the table, and discussion were re-structured to clearly explain the aims of this study. We have re-structured the manuscript to explain, separately, the feasibility, efficacy, and safety data that can be extracted from the selected articles. Specific treatment modalities are subcategorized in separate paragraphs and discussed.
Hyphens, other typos and language errors were corrected. Referrences had some inaccuracies and corrected in the whole manuscript.
Reviewer 2 Report
In this manuscript, the authors provide a review of literature regarding immunotherapy for glioblastoma treatment. This is a relevant topic in modern oncology, and the complexity and volume of data creates an ideal opportunity for a review.
Overall, manuscript organization is poor. Not all points tie in well to the central question of the review. The in-text citations do not all match the reference list. English language editing is also needed. In summary, Major revisions are needed if this manuscript is to be considered.
General Comments:
1) The abstract talks about safety and feasibility of immunotherapy, the introduction talks about efficacy, and the conclusion talks about safety and efficacy. Specify the central question of the review very clearly. In my opinion, it makes the most sense to include all 3 (feasibility, safety, efficacy). Also, how is efficacy measured? Overall survival? Progression-free survival? Once the question is specified, tie each piece of literature into it in the results. In the conclusion, clearly state the answer to the question.
2) More specifics are needed regarding methodology. Out of 750 papers, why were only 50 judged relevant according to the title? This needs to be clearly explained. I know there are other relevant clinical trials that were not included here (i.e., Cloughesy et al., PMID 30742122).
3) The entire results section needs to be re-organized topically. The flow is poor and jumps back and forth between topics. Some suggestions:
Option 1: a) Vaccine-based therapy; b) T-cell based therapy, including CAR-T; c) Immune checkpoint inhibitor therapy
Option 2: a) Therapies with good evidence of survival benefit; b) Therapies with no evidence of survival benefit; c) Therapies with conflicting evidence
Option 3: a) Feasibility; b) Safety; c) Efficacy
4) Cross check each in-text citation and make sure it correlates with the reference list. Specific examples of inconsistencies are noted below.
5) Editing of English language and style is strongly recommended
Specific comments:
1) Introduction, line 43: "Complete surgical removal" should be "Gross total resection."
2) Introduction, line 44: The authors might consider including tumor treating fields (TTF) in GBM standard of care, since this therapy improves survival and is becoming more widely used.
3) Introduction, lines 46-47: Are local recurrence and rare metastases relevant to immunotherapy? If so, please explain how. If not, please delete these comments.
4) Figure 1: "Robusting" is not a verb
5) Introduction, lines 86-87: "Recurrence is the primary cause of therapy failure." This phraseology is not clear. It may make more sense to cite the average time to tumor recurrence post-therapy.
6) Introduction, line 87: What is meant by "Specific treatment restrictions?" Please clarify or re-phrase.
7) Results, lines 130-132. "The studies analyzed...," This entire sentence does not make sense and needs to be re-written.
8) Results, line 134: Reap et al. is reference number 31, not reference number 18.
9) Reference number 19 is not by George et al., and does not discuss Durvalumab. Are the authors referring to PMID 35483906?
10) Results, lines 173-175: If the authors choose to organize results by type of therapy, this would be an ideal place to talk about the Checkmate 498 trial, Durvalumab, and Cloughesy's study of immune checkpoint blockade in hypermutated gliomas (PMID 30742122).
11) Discussion, lines 191-195: Include the findings of Checkmake 498 in the results, not the discussion.
12) Discussion, lines 199-200: This sentence refers to "at least three publications," but has only two references.
13) Conclusions, line 245: What is meant by "these new treatments"? Immunotherapy? If so, please specify.
Author Response
To Reviewer 2
Dear Dr/Prof
Thank you for your very nice comments.
The manuscript was revised entirely. All sections including the introduction, results, table, and discussion were re-structured to clearly explain the aims of this study. We have restructured the manuscript to explain, separately, the feasibility, efficacy, and safety data that can be extracted from the selected articles. Specific treatment modalities are subcategorized in separate paragraphs and discussed.
Hyphens, other typos, and language errors were corrected. Referrences had some inaccuracies and was corrected in the whole manuscript. The first figure was found redundant and not informative enough and so deleted.
The specific issues you mentioned are all corrected point-by-point and you can find them in the manuscript.
thank you for your attention
Round 2
Reviewer 1 Report
The manuscript has been extensively revised and improved. Indeed, the message is more clear, concise and incisive.
Author Response
Dear Reviewer,
we greatly appreciated you comment which allow to improve the quality of the manuscript. Thank you very much for your suggestions.
Reviewer 2 Report
The authors have revised the manuscript substantially in accordance with recommendations. The manuscript is now much improved, but still has some minor issues. With some additional minor revisions.
General comments:
1) When re-submitting a revised manuscript, always provide a point-by-point response to every reviewer comment, and describe exactly how the manuscript has been revised to address the comment. This is the standard expectation in most journals, and is good practice in general. The statement: "The specific issues you mentioned are all corrected point-by-point and you can find them in the manuscript" is not adequate, especially if major revisions are required, as in this case. The authors have addressed all of my critiques, so I am willing to let it go, but take note that some reviewers may reject a revised manuscript outright if it lacks a point-by-point response.
2) The English language and style have been improved, but would still benefit from some additional editing.
Specific comments:
1) Abstract, line 27: “GB” should be “GBM.”
2) Introduction, line 49: A reference is needed for tumor-treating fields (TTF). Use PMID 29260225.
Author Response
When re-submitting a revised manuscript, always provide a point-by-point response to every reviewer comment, and describe exactly how the manuscript has been revised to address the comment. This is the standard expectation in most journals, and is good practice in general. The statement: "The specific issues you mentioned are all corrected point-by-point and you can find them in the manuscript" is not adequate, especially if major revisions are required, as in this case. The authors have addressed all of my critiques, so I am willing to let it go, but take note that some reviewers may reject a revised manuscript outright if it lacks a point-by-point response.
Dear Reviewer,
We are very sorry for the mistake. There were many revisions to be made and the time allotted was very short. Due to this we made such a mistake.
Thank you for understanding. It was a mistake due to time pressure. We apologise.
We also thank you for your comments
Regarding to specific comments below you will find the answers
1)Abstract, line 27: “GB” should be “GBM.”
I corrected it. Thank you
2) Introduction, line 49: A reference is needed for tumor-treating fields (TTF). Use PMID 29260225.
As you suggested we have added the following reference and consequently changed the order of all the references. Thanks
“Stupp, R.; Tailibert, S.; Kanner, A.; Read, W., Steinberg, D.; Lhermitte, B.; Toms, S.; Idbaih, A.; Ahluwalia, M.S.; Fink, C.; Effect of Tumor-Treating Fields Plus Maintenance Temozolomide vs Maintenance Temozolomide Alone on Survival in Patients With Glioblastoma: A Randomized Clinical Trial. JAMA . 2017 Dec 19;318(23):2306-231. doi: 10.1001/jama.2017.18718.”